# Inhibitory Effects of Eicosapentaenoic Acid on Vascular Endothelial Growth Factor-Induced Monocyte Chemoattractant Protein-1, Interleukin-6, and Interleukin-8 in Human Vascular Endothelial Cells

**DOI:** 10.3390/ijms25052749

**Published:** 2024-02-27

**Authors:** Yoko Takenoshita, Akinori Tokito, Michihisa Jougasaki

**Affiliations:** Institute for Clinical Research, NHO Kagoshima Medical Center, Kagoshima 892-0853, Japan; takenoshita.yoko.cj@mail.hosp.go.jp (Y.T.); tokininn@hotmail.com (A.T.)

**Keywords:** eicosapentaenoic acid, vascular endothelial growth factor, monocyte chemoattractant protein-1, mitogen-activated protein kinase, nuclear factor-kappa B, vascular endothelial cells

## Abstract

Vascular endothelial growth factor (VEGF) induces monocyte chemoattractant protein-1 (MCP-1) and plays an important role in vascular inflammation and atherosclerosis. We investigated the mechanisms of VEGF-induced MCP-1 expression and the effects of eicosapentaenoic acid (EPA) in human umbilical vein endothelial cells (HUVECs). Real-time reverse transcription polymerase chain reaction (RT-PCR) and enzyme-linked immunosorbent assay (ELISA) demonstrated that VEGF enhanced MCP-1 gene expression and protein secretion in HUVECs. Western immunoblot analysis revealed that VEGF induced the phosphorylation of p38 mitogen-activated protein kinase (MAPK) and inhibitor of nuclear factor (NF)-κB (IκB). Treatment with pharmacological inhibitors of p38 MAPK (SB203580) or NF-κB (BAY11-7085) significantly suppressed VEGF-induced MCP-1 in HUVECs. EPA inhibited VEGF-induced MCP-1 mRNA, protein secretion, phosphorylation of p38 MAPK, and the translocation of phospho-p65 to the nucleus. Additionally, VEGF also stimulated gene expressions of *interleukin (IL)-6* and *IL-8*, which were suppressed by SB203580, BAY11-7085, and EPA. The present study has demonstrated that VEGF-induced activation of MCP-1, IL-6, and IL-8 involves the p38 MAPK and NF-κB signaling pathways and that EPA inhibits VEGF-induced MCP-1, IL-6, and IL-8 via suppressing these signaling pathways. This study supports EPA as a beneficial anti-inflammatory and anti-atherogenic drug to reduce the VEGF-induced activation of proinflammatory cytokine and chemokines.

## 1. Introduction

Atherosclerosis is recognized as a chronic inflammatory disease of the vessel wall [1]. Neo-angiogenesis is deeply involved in plaque instability and causes consequent plaque rupture. Vascular endothelial growth factor (VEGF), which plays an important role in angiogenesis, causing cell proliferation, apoptosis inhibition, increased vascular permeability, vasodilatation, and recruitment of inflammatory cells to the injury site [2,3,4], is involved in the development of atherosclerosis and furthers cardiovascular diseases [5]. The VEGF signal transduction system involves the phosphoinositide-3-kinase (PI3K)/Akt, p38 mitogen-activated protein kinase (MAPK), and extracellular signal-regulated kinase (ERK) 1/2, and nuclear factor-kappa B (NF-κB) pathways [3,6]. On the other hand, monocyte chemoattractant protein (MCP)-1, also called CC-motif ligand (CCL) 2, is a member of the CC chemokine family and promotes cell migration and infiltration of inflammatory cells like monocytes/macrophages [7]. Accumulating evidence has revealed that VEGF induces MCP-1 in human and bovine vascular endothelial cells [6,8], and MCP-1 induces VEGF in an opposite fashion [9,10], delineating a positive feedback loop between VEGF and MCP-1. In addition, interleukin (IL)-6, a multi-functional proinflammatory cytokine, and IL-8, another chemokine known as CXCL8, are both involved in the pathophysiology of inflammation and atherosclerosis [11,12,13]. However, the precise relationship between these interleukins and VEGF in the pathophysiology of vascular inflammation and atherosclerosis still remains unclarified.

Although statins are prescribed worldwide for atherosclerotic cardiovascular disease, high triglyceride levels may persist in some patients despite statin therapy. Several triglyceride-lowering drugs are available, including fibrates, niacin, and omega-3 polyunsaturated fatty acids, of which prescription omega-3 polyunsaturated fatty acids have the best tolerability and safety profile [14,15]. Eicosapentaenoic acid (EPA) is a long-chain omega-3 polyunsaturated fatty acid that is mainly obtained from marine blue fish. EPA reduces both pro-inflammatory cytokines and chemokines and has been recently used as a drug for hyperlipidemia, preventing atherosclerotic cardiovascular lesions [16]. Matsumoto et al. showed that the administration of EPA suppressed the development of atherosclerotic lesions in a mouse model of hyperlipidemia [17]. The authors also demonstrated that EPA treatment attenuated TNF-α-induced up-regulation of MCP-1 in HUVECs [17]. In addition, Koto et al. reported that EPA treatment resulted in a significant inhibition of MCP-1 in tissue necrosis factor (TNF)-α-stimulated murine vascular endothelial cells and that of VEGF in lipopolysaccharide (LPS)-stimulated murine macrophages [18].

The present study was designed to investigate the signaling pathways involved in the VEGF-induced activation of MCP-1, IL-6, and IL-8 in human umbilical vein endothelial cells (HUVECs). We also elucidated the effects of EPA on the VEGF-induced expressions of MCP-1, IL-6, and IL-8 in HUVECs.

## 2. Results

### 2.1. VEGF-Induced Gene Expression and Protein Secretion of MCP-1 in HUVECs

Real-time reverse transcription polymerase chain reaction (RT-PCR) showed that VEGF significantly enhanced *MCP-1* mRNA expression in HUVECs at 1 to 4 h, peaking at 4 h after stimulating with VEGF and declined at 8 h (Figure 1A). HUVECs treated with VEGF for 4 h expressed *MCP-1* mRNA, with a significant increase at doses of 5 to 20 ng/mL (Figure 1B). Enzyme-linked immunosorbent assay (ELISA) demonstrated that VEGF stimulated the secretion of the MCP-1 protein from HUVECs, peaking at 24 h compared with untreated control (Figure 1C). The MCP-1 protein secretion that was stimulated by treatment with VEGF for 24 h was significantly increased at doses of 5 to 20 ng/mL of VEGF in HUVECs (Figure 1D).

### 2.2. VEGF-Induced Phosphorylation of p38 MAPK and IκB in HUVECs

HUVECs were stimulated by VEGF for different time periods (5–120 min), and the protein extracts were examined by Western immunoblot analysis. VEGF phosphorylated p38 MAPK and inhibitor of NF-κB (IκB), peaking at 5 to 15 min and declining at 60 min (Figure 2A). In addition, VEGF phosphorylated the signaling pathways of p38 MAPK and IκB in a dose-dependent manner (Figure 2B).

### 2.3. Effects of Pharmacological Inhibitors of the p38 MAPK and NF-κB Signaling Pathways on VEGF-Induced Gene Expression and Protein Secretion of MCP-1 in HUVECs

To examine whether the p38 MAPK and NF-κB signaling pathways are involved in the VEGF-induced gene expression and protein secretion of MCP-1, SB203580 (p38 MAPK inhibitor) and Bay11-7085 (NF-κB inhibitor) were used, followed by stimulation with VEGF. As shown in Figure 3A, the VEGF-induced gene expression of *MCP-1* was significantly inhibited by pretreatment with SB203580 and Bay11-7085. Similarly, the pharmacological inhibitors SB203580 and Bay11-7085 suppressed the VEGF-induced protein secretion of MCP-1 from HUVECs (Figure 3B).

### 2.4. Effects of EPA on Cell Viability

The cytotoxicity of various doses of EPA to the cultured HUVECs was examined using the MTT assay. The cell viability of HUVECs was not changed by treatment with EPA at doses of less than 100 μmol/L (Figure 4). Although no changes in cell viability were observed in HUVECs at a dose of 100 μmol/L of EPA, we used EPA at doses of 10 and 30 μmol/L instead of 100 μmol/L in the present experiments, considering the concentration used in the previous studies [19].

### 2.5. Effects of EPA on the VEGF-Induced Gene Expression and Protein Secretion of MCP-1 in HUVECs

To elucidate the effects of EPA on the VEGF-induced gene expression and protein secretion of MCP-1, HUVECs were pretreated with various concentrations of EPA (10 and 30 μmol/L) overnight, followed by stimulation with VEGF for 4 h to examine *MCP-1* gene expression in HUVECs and for 24 h to measure MCP-1 protein secretion from HUVECs. The treatment with EPA (10 and 30 μmol/L) significantly inhibited the VEGF-induced increase in *MCP-1* gene expression in HUVECs (Figure 5A). On the other hand, the VEGF-stimulated increase in MCP-1 protein secretion from HUVECs was significantly suppressed only at the dose of 30 μmol/L of EPA (Figure 5B).

### 2.6. Effects of SB203580, BAY11-7085, and EPA on the VEGF-Induced Gene Expression of IL-6 and IL-8 in HUVECs

Real-time PCR demonstrated that VEGF significantly increased the gene expression of *IL-6* and *IL-8* in HUVECs. SB203580 (p38 MAPK inhibitor), BAY11-7085 (NF-κB inhibitor), and EPA significantly inhibited the VEGF-stimulated gene expression of *IL-6* (Figure 6A) and *IL-8* (Figure 6B) in HUVECs.

### 2.7. Effects of EPA on the VEGF-Stimulated Phosphorylation of p38 MAPK in HUVECs

To investigate whether the VEGF-induced phosphorylation of p38 MAPK was suppressed by EPA, HUVECs were pretreated with various concentrations of EPA (10 and 30 μmol/L) overnight, followed by stimulation with VEGF for 5 min. Western immunoblot analysis revealed that EPA slightly but significantly inhibited the VEGF-induced phosphorylation of p38 MAPK (Figure 7).

### 2.8. Immunofluorescence Staining

Immunofluorescence staining was used to examine whether EPA affects the translocation of phospho-p65 to the nucleus by suppressing VEGF-stimulated p65 phosphorylation. HUVECs were pretreated with EPA or Bay11-7085, followed by treatment with VEGF for 60 min. The immunofluorescence signal of phospho-p65 was localized in the nuclei of HUVECs after incubation with VEGF for 60 min compared with untreated control cells. VEGF-induced phospho-p65 activation was inhibited by EPA at the dose of 30 μmol/L (Figure 8). Similarly, Bay11-7085 attenuated the VEGF-induced translocation of phospho-p65 to the nucleus.

## 3. Discussion

Atherosclerosis is characterized by chronic inflammation of the vessel wall [1]. Neo-angiogenesis in atherosclerotic plaques is associated with unstable plaque formation and consequent rupture risk. VEGF plays a pivotal role in angiogenesis, causing recruitment of inflammatory cells to the injury site, and is involved in the development of atherosclerosis and furthers cardiovascular diseases [5]. EPA, a representative of the omega-3 polyunsaturated fatty acids, is known to reduce plaque instability and plaque inflammation [20]. The current study has demonstrated that EPA significantly inhibited the VEGF-stimulated activation of proinflammatory cytokines and chemokines in the vascular endothelial cells and provides new insights into the roles of EPA in the pathophysiology of vascular inflammation and atherosclerosis.

MCP-1 and IL-8 are the chemokines that recruit monocytes/macrophages and neutrophils to the sites of action, respectively. MCP-1 is a member of the CC class of chemokine supergene family, whereas IL-8 is of the CXC class, both of which play important roles in the inflammatory diseases [11] and are implicated in atherogenesis [12]. On the other hand, IL-6 is a multi-functional proinflammatory cytokine that is involved in immune regulation, inflammation, metabolism, and tissue regeneration [21,22]. Recently, a causal role for IL-6 in systemic atherothrombosis and aneurysm formation and the potential role of IL-6 inhibition in stable coronary disease, acute coronary syndromes, heart failure, and the atherothrombotic complications associated with chronic kidney disease and end-stage renal failure have been reported [13]. In the present study, EPA effectively suppressed the VEGF-induced activation of chemokines (MCP-1 and IL-8), as well as a proinflammatory cytokine (IL-6), in the vascular endothelial cells. EPA inhibits monocyte recruitment to the atherosclerotic lesions and subsequent conversion to macrophages and foam cells, reducing atherosclerotic plaque formation and the vulnerability to rupture [17,20]. These beneficial actions of EPA may be related to the suppressing effects of EPA on the VEGF-induced activation of chemokines (MCP-1 and IL-8) and a proinflammatory cytokine (IL-6), as shown in the present study.

The current study showed that VEGF stimulated MCP-1 gene expression and protein secretion in the human vascular endothelial cells. We sought to investigate the signal transduction pathways involved in the VEGF-induced MCP-1 activation and found that VEGF-induced MCP-1 activation occurred through activation of the p38 MAPK and NF-κB pathways in the human vascular endothelial cells. The finding that VEGF induces MCP-1 expression in the vascular endothelium is supported by previous studies [6,8]. Marumo et al. demonstrated that VEGF induced the gene expression and protein secretion of MCP-1 via the activation of NF-κB and activator protein (AP)-1 binding activity in bovine retinal endothelial cells, suggesting an important role for VEGF-induced MCP-1 in the development of microvascular angiopathy. Using the selective ERK1/2 inhibitor PD98059, the authors also found that induction of MCP-1 expression by VEGF was also dependent on the ERK1/2 pathway [6]. Yamada et al. revealed that MCP-1 was an important factor in the angiogenesis process and the vascular leakage induced by VEGF and that AP-1 was directly involved in the VEGF-induced MCP-1 expression in the vascular endothelium [8]. On the other hand, several investigations have reported MCP-1-induced VEGF activation in the literature. Parenti et al. showed that MCP-1 increased the expressions of *VEGF* mRNA and protein and that MCP-1 stimulated proliferation and migration of rat vascular smooth muscle cells through the activation of endogenous VEGF [10]. Other investigators have demonstrated that MCP-1 stimulated VEGF production through the activation of ERK1/2 in human aortic endothelial cells [9]. These findings reveal that angiogenic VEGF induces chemotactic MCP-1 expression and that MCP-1 also induces VEGF in an opposite fashion, delineating a positive feedback regulatory loop between VEGF and MCP-1 in the vascular endothelium. Crosstalk between VEGF and MCP-1 is intriguing and needs further investigation.

EPA, a representative of the omega-3 polyunsaturated fatty acids, has been clinically prescribed in patients with hyperlipidemia. EPA has a variety of pharmacological properties including lowering triglycerides [23], improvement of endothelial function via nitric oxide production [24], vasodilatation [24,25], and anti-inflammatory actions [26]. MCP-1 is an important chemokine that plays a crucial role in pathological conditions, such as cardiovascular diseases including atherosclerosis, brain pathologies, bone and joint disorders, respiratory infections, endothelial dysfunction, and cancer [7]. In this study, EPA treatment effectively inhibited VEGF-mediated MCP-1 induction by suppressing the signal transduction systems of the p38 MAPK and NF-κB signaling pathways. Previous studies also investigated the inhibitory properties of EPA on MCP-1 expression. Matsumoto et al. conducted a study on a mouse model of hyperlipidemia and demonstrated that the administration of EPA reduced the development of atherosclerotic lesions in this animal model [17]. The atherosclerotic plaques of EPA-fed mice revealed a stable morphology in association with a lower deposition of lipids and a reduced accumulation of macrophages and an increase in smooth muscle cells and collagen content. In addition, EPA treatment attenuated the TNF-α-induced up-regulation of adhesion molecules and MCP-1 in HUVECs [17]. EPA effectively decreased LPS-induced NF-κB activation and MCP-1 expression in human proximal tubular cells [27]. Akekura et al. demonstrated that EPA inhibited the LPS-induced phosphorylation of the NF-κB p65 subunit in a mouse monocyte/macrophage cell line [28]. EPA partially but significantly suppressed LPS-induced MCP-1 gene expression in these cells in vitro. The authors also showed that MCP-1 expression was induced in the adventitia of intracranial aneurysm and that its expression was remarkably suppressed in the intracranial aneurysm lesions from EPA-treated rats [28]. EPA suppressed TNF-α-stimulated MCP-1 transcription by preventing NF-κB activation in an ERK-dependent fashion in the cultured rat mesangial cells [29]. In addition, Koto et al. performed in vitro experiments and showed that EPA treatment led to a significant inhibition of the mRNA expression and protein levels of MCP-1 in TNF-α-stimulated murine brain-derived capillary endothelial cells and decreased *VEGF* mRNA and protein secretion in LPS-stimulated murine macrophages [18].

Other investigators have also reported on the biological properties of EPA to suppress VEGF expression. Yang et al. demonstrated that treatment with EPA for 48 h resulted in a dose-dependent suppression of VEGF-induced proliferation in bovine carotid artery endothelial cells [30]. VEGF-activated MAPK was also inhibited by treatment with EPA in these cells [30]. Serum-starvation-induced constitutive VEGF expression was reduced by treatment with EPA through inhibition of the ERK1/2 signaling pathway in human colon cancer cells [19]. EPA dose-dependently suppressed cell proliferation and wound repair in cultured human microvascular endothelial cells, and EPA significantly suppressed the gene expression and protein secretion of VEGF in both normoxia and hypoxia culture conditions [31]. Tevar et al. investigated the effects of supplemental dietary EPA in an animal model of progressive malignancy and found that EPA supplementation inhibited tumor growth, potentially through alterations in the expression of the pro-angiogenic VEGF [32]. These findings demonstrate that EPA inhibits MCP-1 and/or VEGF and might play an important role in the underlying mechanisms of the beneficial effects in the treatment of vascular inflammation and atherosclerosis. Further studies are required to investigate the role of EPA in vascular inflammation and atherosclerosis.

Although the present study indicates a plausible role for EPA in the VEGF-induced activations of MCP-1, IL-6, and IL-8 in vascular endothelial cells, it has some limitations. First, we used HUVECs as representative vascular endothelial cells in the current study. However, there is significant endothelial cell phenotype heterogeneity across the vascular tree [33]. At present, no single endothelial cell line is representative of the endothelium in all blood vessels [34] and HUVECs are commonly used in in vitro experiments for endothelial-derived gene expression and the protein secretion of cytokines, such as IL-8 [35], GRO-α [36], and MCP-1 [37]. Second, the current study is an in vitro cell culture study and does not involve preclinical or clinical data. Inflammation is central to the pathophysiology of atherosclerosis. The present study demonstrated that EPA suppressed the VEGF-induced activation of a proinflammatory cytokine (IL-6) and some chemokines (MCP-1 and IL-8), delineating a possible role for EPA in the treatment of atherosclerosis due to anti-inflammatory properties. Third, secretion studies of IL-6 and IL-8 are lacking because of the depletion of the supernatant samples. However, it is likely that the secretion pattern of IL-6 and IL-8 resembles that of MCP-1, because the gene expression of these interleukins is similar to that of MCP-1. Further studies using ELISA are needed to confirm the secretion of IL-6 and IL-8 in HUVECs stimulated by VEGF in the presence and absence of EPA.

## 4. Materials and Methods

### 4.1. Regents

Recombinant human VEGF was obtained from PeproTech (Rocky Hill, NJ, USA). The rabbit polyclonal antibodies for p38, phospho-p38 (Thr180/Tyr182), IκB, phospho-IκB (Ser32/36), and phospho-p65 (Ser536) were obtained from Cell Signaling Technology (Beverly, MA, USA). The rabbit polyclonal anti-GAPDH antibody was from Santa Cruz Biotechnology (Heidelberg, Germany). Pharmacological inhibitors SB203580 (p38 MAPK inhibitor) and BAY11-7085 (NF-κB inhibitor) were purchased from FUJIFILM Wako Pure Chemical (Osaka, Japan) and Cayman Chemical (Ann Arbor, MI, USA), respectively. EPA was purchased from Merck (Darmstadt, Germany).

### 4.2. Cell Culture of HUVECs

HUVECs were purchased from Kurabo (Osaka, Japan), seeded in plastic plates pre-coated with type I collagen (Asahi Techno Glass, Nagoya, Japan), and maintained in medium 199 (Life Technologies, Carsbad, CA, USA) supplemented with 10% heat-inactivated fetal calf serum, 0.5 μg/mL fungizone, 0.25 μg/mL amphotericin B, 100 μg/mL streptomycin, 100 U/mL penicillin (all Life Technologies, Carlsbad, CA, USA), 14 U/mL heparin (Ajinomoto, Tokyo, Japan), 20 μg/mL endothelial cell growth supplement (Kohjin Bio, Saitama, Japan), and 10 μg/mL human epidermal growth factor (PeproTech, Rocky Hill, CT, USA). HUVECs were cultured at 37 °C in 5% CO_2_ and 95% air in a humidified atmosphere.

### 4.3. Total RNA Extraction and Real-Time RT-PCR

Total RNA was extracted from HUVECs using the Pure Link RNA Mini Kit (Invitrogen, Carlsbad, CA, USA). The cDNA was synthesized using the Superscript VILO cDNA Synthesis kit (Invitrogen, Carlsbad, CA, USA). Real-time PCR using Power SYBR Green PCR Master Mix (Applied Biosystems, Warrington, UK) was carried out on a CFX connect thermal cycler (Bio-Rad, Hercules, CA, USA). The value of each cDNA was calculated using the ΔΔCq method and normalized to the value of the housekeeping gene *GAPDH*.

Oligonucleotide PCR primers targeting human *MCP-1* mRNA were designed according to a previous report [37] and primers targeting human *GAPDH* mRNA were purchased from TaKaRa (Shiga, Japan); the specificity of the primer sets was verified by a basic local alignment search tool (BLAST) search and melting-curve analysis. The primer sequences and accession numbers are shown in Table 1. The reaction conditions were as follows: an activation step at 95 °C for 10 min, followed by 40 cycles of denaturation at 95 °C for 15 s and annealing/extension at 60 °C for 1 min.

### 4.4. Enzyme-Linked Immunosorbent Assay (ELISA)

The concentrations of MCP-1 in the culture medium were determined by using a human MCP-1 ELISA kit (R&D Systems, Minneapolis, MN, USA) according to the manufacturer’s protocol. The optical densities of the samples and standards were measured spectrophotometrically with an iMark microplate reader (BIORAD, Hercules, CA, USA). MCP-1 concentrations were evaluated by comparing the optical density with the standard curve.

### 4.5. Western Immunoblot Analysis

Western immunoblot analysis was performed as previously reported, with some modifications. HUVECs were harvested in ice-cold cell lysis buffer together with phenylmethylsulphonyl fluoride and a protease inhibitor cocktail. The proteins were resuspended in sodium dodecyl sulfate sample buffer and dithiothreitol, sonicated, and boiled for 5 min. They were separated by 4–12% NuPAGE Bis-Tris gels (Life Technologies, Carlsbad, CA, USA) and transferred to a polyvinylidene difluoride membrane with a Trans-Blot Turbo Transfer System (Bio-Rad, Hercules, CA, USA) for 7 min. The membrane was soaked in a 5% nonfat dry milk blocking buffer. The membrane was then incubated with the primary antibody overnight at 4 °C in the concentrations suggested by the manufacturer, followed by 1 h incubation with horseradish-peroxidase-conjugated secondary antibody (Cell Signaling Technology, Beverly, MA, USA). ECL prime (GE Healthcare, Buckinghamshire, UK) was used to visualize the protein bands, and the intensities of the blots were quantified by a ChemiDoc Touch Imaging System (Bio-Rad, Hercules, CA, USA).

### 4.6. Cell Viability

Cell viability was assessed using the MTT assay (Roche, Mannheim, Germany). HUVECs were treated with 10, 30, 100, or 300 μmol/L of EPA overnight, after which 0.5 mg/mL MTT solution was added to the culture medium and then the samples were incubated for 4 h. After they were incubated with dimethyl sulfoxide overnight, the cell viability was measured with an iMark microplate reader (BIORAD, Hercules, CA, USA). The survival rates of the EPA-treated cells were compared with those of the untreated control cells.

### 4.7. Immunofluorescence Staining

After fixation with 1% paraformaldehyde, HUVECs were permeabilized with 0.1% Triton X-100 and blocked with normal horse serum for 30 min. The cells were incubated with a rabbit p65 antibody at a 100-fold dilution overnight. They were then washed and incubated with anti-rabbit IgG-Alexa (Cell Signaling Technology, Beverly, MA, USA) at a dilution of 250-fold for 1 h, and the nuclei were counterstained using Hoechst 33342 (Invitrogen, Carlsbad, CA, USA). Images were analyzed using a fluorescence microscope (Olympus, Tokyo, Japan).

### 4.8. Statistical Analysis

Data are shown as mean ± SD. The average of three to eight independent experiments were represented in each data point. The statistical significance of the data was assessed using one-way ANOVA with Tukey–Kramer’s post hoc test. A *p* value < 0.05 was considered statistically significant.

## 5. Conclusions

The present study has demonstrated that VEGF induces MCP-1 expression via the p38 MAPK and NF-κB signaling pathways and that EPA inhibits these signaling pathways and suppresses VEGF-induced MCP-1 expression in the human vascular endothelial cells. This study supports the properties of EPA as a beneficial anti-inflammatory and anti-atherogenic drug in the treatment of vascular inflammation and atherosclerosis.

## Figures and Tables

**Figure 1 ijms-25-02749-f001:**
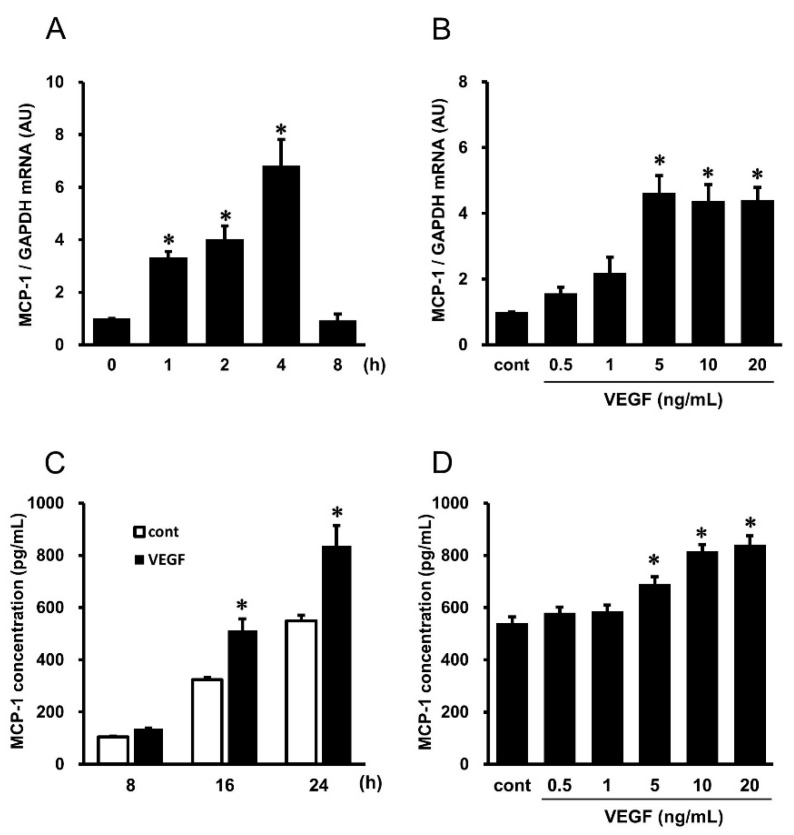
Vascular endothelial growth factor (VEGF)-stimulated gene expression and protein secretion of monocyte chemoattractant protein (MCP)-1 in human umbilical vein endothelial cells (HUVECs). (**A**) Time course of *MCP-1* mRNA expression after treatment with 10 ng/mL of VEGF (*n* = 3), as evaluated by real-time reverse transcription polymerase chain reaction (RT-PCR). (**B**) *MCP-1* mRNA expression in HUVECs after treatment with the indicated concentrations of VEGF for 4 h (*n* = 3), as evaluated by real-time RT-PCR. Bars represent *MCP-1* mRNA after normalization to *glyceraldehyde-3-phosphate dehydrogenase* (*GAPDH*) mRNA and relative to 0 h in (**A**) and untreated control (cont) in (**B**). (**C**) Time course of MCP-1 concentrations in the culture supernatant after treatment with 10 ng/mL of VEGF (closed bars, *n* = 6), as analyzed by ELISA. The control, secretion of MCP-1 without VEGF treatment, is shown in open bars (*n* = 6), as analyzed by ELISA. (**D**) MCP-1 concentrations in the culture supernatant after treatment with the indicated concentrations of VEGF for 24 h (*n* = 6), as analyzed by ELISA. * *p* < 0.05 vs. 0 h in (**A**), vs. untreated control (cont) in (**B**,**D**), and vs. each control at the same time in (**C**).

**Figure 2 ijms-25-02749-f002:**
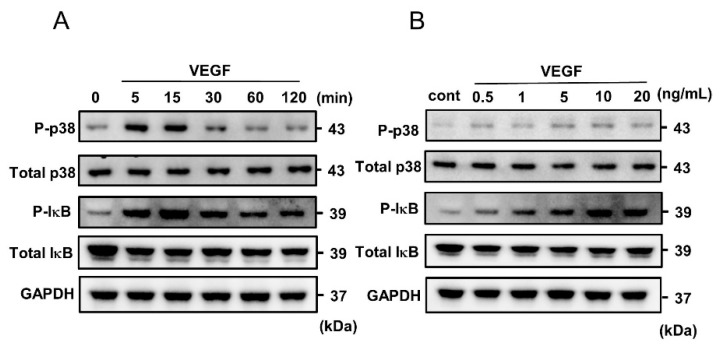
Western immunoblot analysis showing VEGF-stimulated phosphorylation of p38 mitogen-activated protein kinase (MAPK) and inhibitor of nuclear factor (NF)-kappa B (IκB) in HUVECs. (**A**) HUVECs were treated with 10 ng/mL of VEGF for 5, 15, 30, 60, and 120 min. (**B**) HUVECs were treated with different concentrations of VEGF (0.5, 1, 5, 10, and 20 ng/mL) for 5 min.

**Figure 3 ijms-25-02749-f003:**
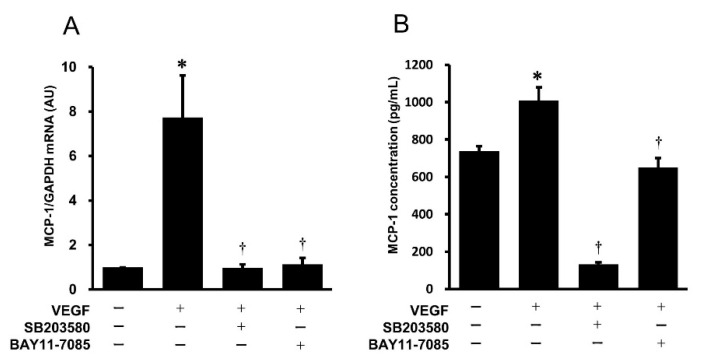
Effects of pharmacological inhibitors of the p38 MAPK and NF-κB pathways on VEGF-induced gene expression (**A**) and protein secretion (**B**) of MCP-1 in HUVECs. HUVECs were preincubated with SB203580 (10 μmol/L) and BAY11-7085 (10 μmol/L) for 2 h, followed by stimulation with VEGF (10 ng/mL) for 4 h to examine *MCP-1* mRNA (**A**) or for 24 h to measure MCP-1 protein concentration (**B**). *MCP-1* mRNA was evaluated by real-time RT-PCR ((**A**), *n* = 3), and MCP-1 concentration was examined by ELISA ((**B**), *n* = 6). * *p* <0.05 vs. untreated control. † *p* <0.05 vs. VEGF.

**Figure 4 ijms-25-02749-f004:**
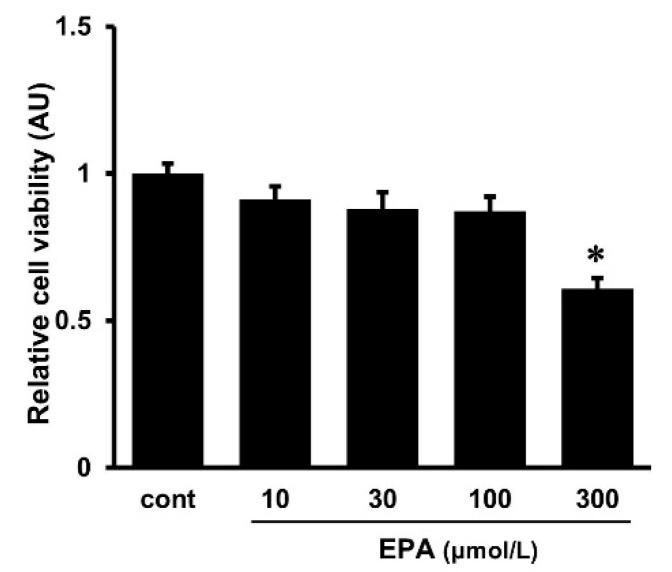
HUVEC viability when subjected to different concentrations (10, 30, 100, 300 μmol/L) of eicosapentaenoic acid (EPA) for 24 h. Cell viability was measured using the MTT assay. The results are expressed as a percentage of the untreated control (cont), and each value represents eight independent experiments (*n* = 8). * *p* < 0.05 vs. untreated control (cont).

**Figure 5 ijms-25-02749-f005:**
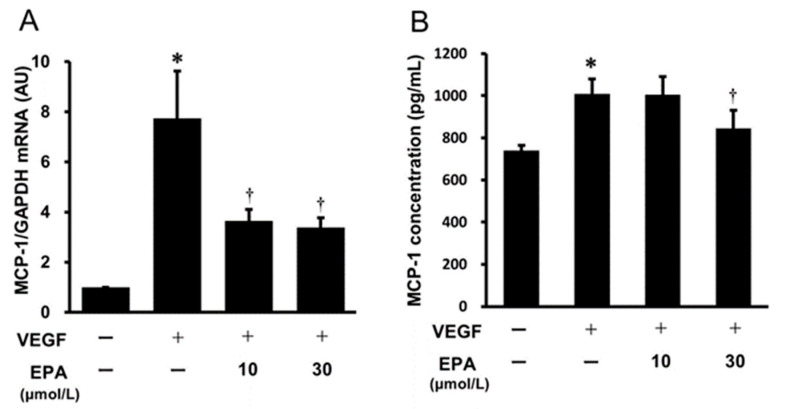
Effects of EPA on the gene expression and protein secretion of MCP-1 in HUVECs. VEGF-induced *MCP-1* mRNA (**A**) and MCP-1 protein concentration (**B**) were suppressed by EPA. HUVECs were treated with VEGF (10 ng/mL) for 4 h (**A**) or 24 h (**B**) with or without pretreatment with EPA (10 and 30 μmol/L). Bars represent *MCP-1* mRNA after normalization to *GAPDH* mRNA and relative to the untreated control in (**A**). Bars represent MCP-1 protein concentrations in (**B**). * *p* < 0.05 vs. untreated control. † *p* < 0.05 vs. VEGF.

**Figure 6 ijms-25-02749-f006:**
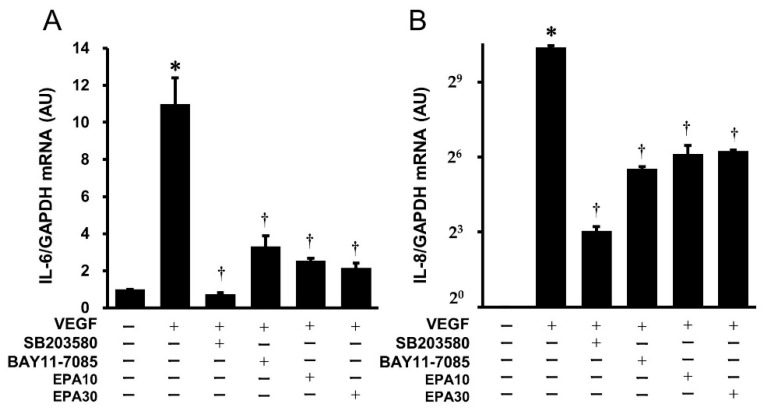
Effects of SB203580, BAY11-7085, and EPA on the VEGF-stimulated gene expression of *IL-6* mRNA (**A**) and *IL-8* mRNA (**B**) in HUVECs. HUVECs were preincubated with SB203580 (10 μmol/L) or BAY11-7085 (10 μmol/L) for 2 h each and EPA (10 and 30 μmol/L) overnight, then stimulated using VEGF (10 ng/mL) for 4 h to examine gene expression of *IL-6* and *IL-8*. Bars represent *IL-6* mRNA (**A**) and *IL-8* mRNA (**B**) after normalization to *GAPDH* mRNA and relative to the untreated control. * *p* < 0.05 vs. untreated control. † *p* < 0.05 vs. VEGF.

**Figure 7 ijms-25-02749-f007:**
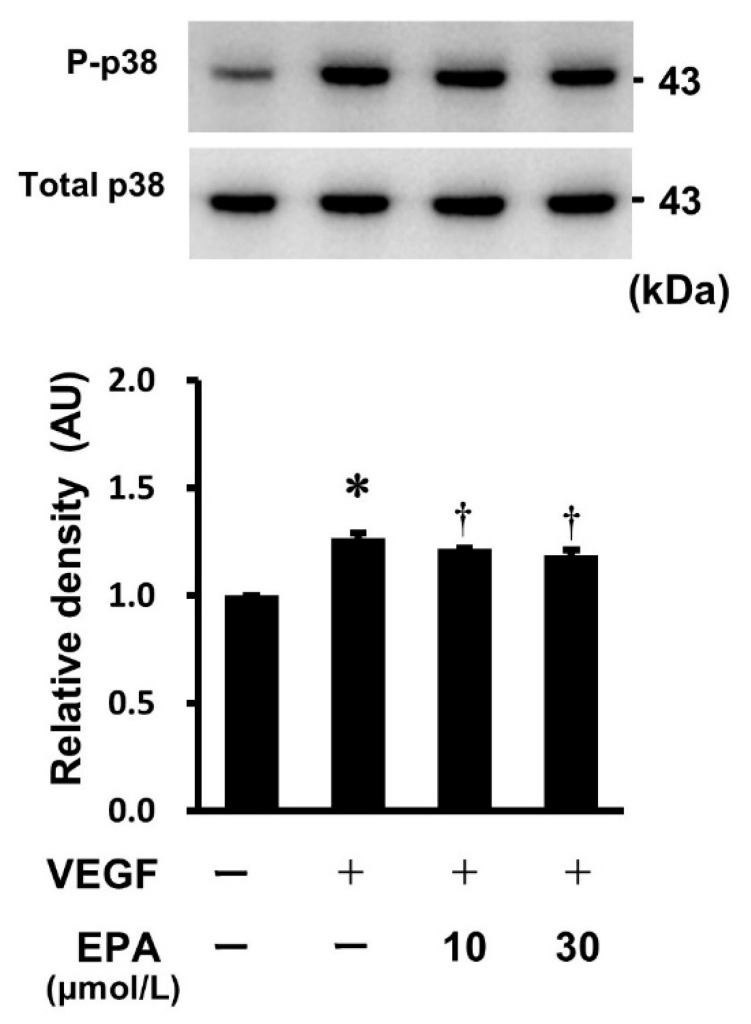
Effects of EPA on the VEGF-induced phosphorylation of p38 MAPK. EPA suppressed the phosphorylation of p38 MAPK in HUVECs. HUVECs were pretreated with EPA (10 and 30 μmol/L) overnight, then incubated with VEGF (10 ng/mL) for 5 min. Bars represent the results from the densitometric analyses of each phosphorylation signal after normalization to total protein and relative to the untreated control. Blots are representative of three independent experiments. * *p* < 0.05 vs. untreated control. † *p* < 0.05 vs. VEGF.

**Figure 8 ijms-25-02749-f008:**
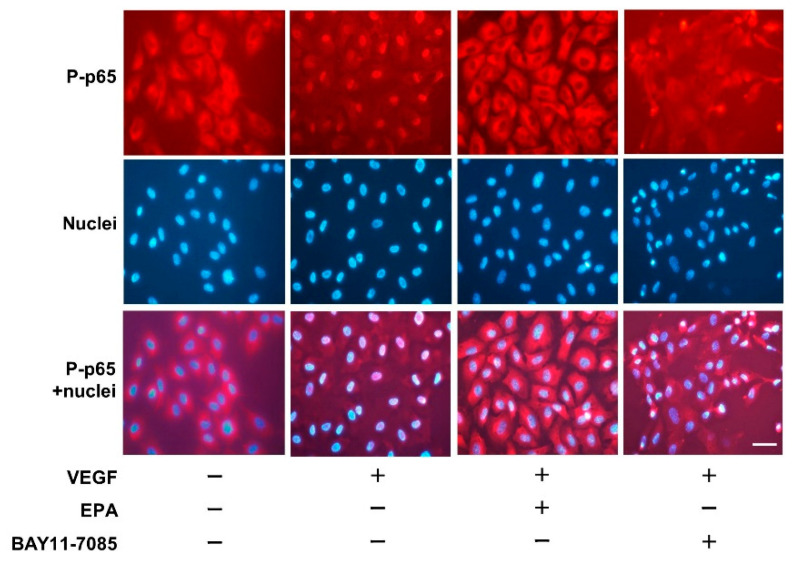
Effects of EPA and BAY11-7085 on VEGF-induced translocation of phospho-p65 to the nucleus, as determined by immunofluorescence staining. HUVECs were pretreated with EPA (30 μmol/L) or BAY11-7085 (10 μmol/L), followed by an additional incubation with VEGF (10 ng/mL) for 60 min. Representative immunofluorescence image showing the localization of phospho-p65 in HUVECs. Red staining indicates the specific Alexa staining for phospho-p65, and blue staining indicates the nuclei (Hoechst 33342). Original magnification, ×400. Scale bar = 50 μm.

**Table 1 ijms-25-02749-t001:** Primers for real-time RT-PCR.

Gene Name	Primer Sequences (Forward/Reverse)	Accession Number
*MCP-1*	F: 5′-CATAGCAGCCACCTTCATTCC-3′ R: 5′-TCTCCTTGGCCACAATGGTC-3′	NM_002982.3
*IL-6*	F: 5′-ACTCACCTCTTCAGAACGAATTG-3′ R: 5′-CCATCTTTGGAAGGTTCAGGTTG-3′	NM_000600.3
*IL-8*	F: 5′-AAGAAACCACCGGAAGGAAC-3′ R: 5′-ACTCCTTGGCAAAACTGCAC-3′	NM_000584.3
*GAPDH*	F: 5′-GCACCGTCAAGGCTGAGAAC-3′ R: 5′-TGGTGAAGACGCCAGTGGA-3′	NM_002046

## Data Availability

The data that support the findings of this study are available from the authors upon reasonable request.

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
