# Peer review of "Inhibitory Effects of Eicosapentaenoic Acid on Vascular Endothelial Growth Factor-Induced Monocyte Chemoattractant Protein-1, Interleukin-6, and Interleukin-8 in Human Vascular Endothelial Cells"

_ijms, 2024, doi:10.3390/ijms25052749_

Round 1

Reviewer 1 Report

Comments and Suggestions for Authors

Major Comments:

1. The abstract is informative and well structured. However, it would benefit if authors could include the number of replicates for each experiment and highlight the novelty of the study.

2. The introduction is very short and should be improved with the incorporation of logical data. Authors should focus on the specific objectives and novelty of the study.The safety profile of the drug (Eicosapentaenoic Acid) should be discussed.

3. In the introduction, authors are suggested to provide a brief explanation of why investigating the role of VEGF-induced MCP-1 expression is relevant to the broader context of vascular inflammation and atherosclerosis.

4. Authors should specify the number of independent experiments conducted for each assay to assess the reliability of the results.

5. In the "Materials and Methods" section, authors describe the concentrations of EPA used in the experiments, including the rationale for selecting these concentrations.  Authors should provide information on the solvent used for EPA and its potential impact on the experimental outcomes.

6. In the discussion section, authors should elaborate on the clinical implications of findings. How might the inhibition of VEGF-induced MCP-1 by EPA translate to potential therapeutic strategies for vascular inflammation and atherosclerosis?

7. Limitation should be discussed in limitation section.

8. Authors should describe the Future perspective and clinical significance of the study.

General Comments:

1. Authors are suggested to ensure that all references are properly cited in the manuscript.

2. Authors should add abbreviation list in the manuscript.

3. Manuscript has many grammatical and spelling mistakes throughout. Authors should ensure consistency in formatting, use of hyphens and proper punctuation throughout the manuscript.

Comments on the Quality of English Language

Manuscript has many grammatical and spelling mistakes throughout. 

Reviewer 2 Report

Comments and Suggestions for Authors

The study describes VEGF-induced MCP1 that is inhibited by EPA. Induction requires p38MAPK and NFkB signalling. Only in vitro data are presented.

The study is limited in scope and must be expanded. The authors could with ease extend their study to include mRNA expression for a number of additional cytokines (IL1, IL6, IL8 and more) using the RNA already at hand from the experiments done.

Figure 4 does not require a separate figure but could be written in the text.

Fig 6 is not convincing. The effects are tiny and the errors bars not visible on three independent experiments. This simply is not realistic considering errors in other experiments.

Comments on the Quality of English Language

Is OK and readable.

Reviewer 3 Report

Comments and Suggestions for Authors

This study showed the involvement of p38 MAPK and NF-κB signaling pathways in VEGF-induced MCP-1 activation and the inhibitory effects of eicosapentaenoic acid on VEGF-induced MCP-1 expression in HUVECs via suppressing the above-mentioned signaling pathways. The results of this work definitely have a value and potential for application in clinical practice. The article is written in a concise manner. After making all corrections and addressing the reviewer’s comments, the manuscript can be published.

Major issue:

According to the title ‘Inhibitory Effects of Eicosapentaenoic Acid on Vascular Endothelial Growth Factor-Induced Monocyte Chemoattractant Protein-1 in Human Vascular Endothelial Cells’, the authors declare about inhibitory and not preventive effects of EPA, while in all their experiments they used pretreatment with this substance and not post-treatment. In my opinion, the chemical may inhibit the effects induced by another factor after the induction is performed but not before that. Even within the figures, you draw the inhibitors’ names (followed by + or – signs) under VEGF but not vice-versa, which looks like you mean that you treated HUVECs with the substances after VEGF treatment (but you actually did it opposite). Please explain. 

Minor issues:

1) ‘Accumulating evidence has revealed that VEGF induces MCP-1 in the human and bovine vascular endothelial cells [3, 5], and vice versa’ (lines 39-40): it is not clear what do you mean by vice versa – that not only VEGF induces MCP-1 but also MCP-1 induces VEGF?  Please explain it somehow here (your explanation of that in the lines 186-188 is far away in the text).

2) Line 56: I would suggest substituting ‘enhanced MCP-1 mRNA’ with ‘enhanced MCP-1 mRNA expression’.

3) Figure 1 (A and B): ‘MCP-1’ and ‘GAPDH’ should be italicized since you are talking about the genes.

4) Figure 1 (C):

(a) Did you measure secretion of MCP-1 later than 24h after treatment with 10 ng/mL of VEGF?

(b) Why did you use exactly this VEGF concentration (I do understand that it is in between 5 and 20, but anyway)?

(c) Why secretion of MCP-1 without VEGF treatment (Control shown in open bars) is ascending upon the time (from 8 to 24 hr)? How could you explain that?

5) Figure 2: when analyzing Western immunoblot data, did you perform a densitometry analysis of the bands (normalizing to total protein levels) in order to make definite conclusions, not only ‘peaking at 5 to 15 min and declined at 60 min’ (though what is written is true)?

6) Lines 90-92 and Figure 3: Why did you use pre-treatment with SB203580 and BAY11-7085 but not post-treatment after induction with VEGF? Could you prove it with the literature data?

7) Figure 3 (A): ‘MCP-1’ and ‘GAPDH’ should be italicized since you are talking about the genes. Additionally, I recommend checking other cases of necessary italicizing throughout the text (like lines 221 and 231 for VEGF, line 268 for MCP-1 and etc.).

8) Similarly, in the subchapter 2.5, Lines 121-122 and Figure 5: Why did you use pre-treatment with EPA but not post-treatment after induction with VEGF? Could you prove it with the literature data?

9) Figure 5 (A): ‘MCP-1’ and ‘GAPDH’ should be italicized since you are talking about the genes.

10) Similarly, in the subchapters 2.6 and 2.7, Lines 137-138 and 152-153, and Figures 6-7, you used pretreatment with EPA. ?

11) The phrase ‘MCP-1 produced a novel transcription factor’ (line 186) sounds a bit awkward (like protein produced protein). Did you mean that MCP-1 triggered a novel transcription factor production?

Round 2

Reviewer 1 Report

Comments and Suggestions for Authors

Authors have diligently addressed almost all the comments and concerns raised during the review process. The revisions made have significantly improved the quality and clarity of the article. However, The iThenticate document viewer has detected a similarity index exceeding the accepted limit (59%). The article in present form is plagiarized with over 20% of its sentences sourced from a single origin (the internet). Authors must re-evaluate the content of the article, ensuring that the similarity index falls within the accepted limit (below 20%).

Reviewer 2 Report

Comments and Suggestions for Authors

no further comments

Author Response

We thank the Reviewer 2 for the efforts to review our manuscript. Although this time the Reviewer 2 did not give us any comments, we are sure that the Reviewer 2 is satisfied with our additional experiment of gene expression of IL-6 and IL-8. Fortunately, we could obtain a novel finding that VEGF induced not only MCP-1, but also IL-6 and IL-8 in the vascular endothelial cells, and that EPA inhibited these inductions provoked by VEGF. Without the Reviewer 2’s comments, we could not investigate these issues. We thank for the Reviewer 2 again for the excellent questions at the highest level.

Reviewer 3 Report

Comments and Suggestions for Authors

Despite the fact that the authors provided less-than-exhaustive responses to my remarks and comments, this does not detract from the overall value of this article. The manuscript may be accepted for publication.

Author Response

We thank the Reviewer 3 for the thoughtful comments. We are encouraged by the comments. We would like to do our best to do more research and write papers.